# Possibilities of Physical-Chemical Sensors’ Use for Optimizing the Processing of Metallurgical Melting Based on Computer Systems

**DOI:** 10.3390/s23083965

**Published:** 2023-04-13

**Authors:** Adrian Ioana, Nicolae Constantin, Alexandra Istrate, Lucian Paunescu, Vili Pasare

**Affiliations:** 1Engineering and Management of Metallic Materials Obtaining Department, Science and Engineering Materials Faculty, University Politehnica of Bucharest, 060042 București, Romania; 2Materials Engineering Department, Science and Engineering Materials Faculty, University Politehnica of Bucharest, 060042 București, Romania; 3Cosfel Actual SRL Bucharest, 021619 Bucharest, Romania

**Keywords:** physical-chemical sensors, computer systems, metallurgical melts, elaboration, quality

## Abstract

This article presents aspects regarding the possibilities of optimizing the quality of the elaboration of metallurgical melts by determining their physical-chemical properties. Thus, the article analyzes and presents methods for determining the viscosity and electrical conductivity related to metallurgical melts. Among the viscosity determination methods, two methods are presented, namely: the rotary viscometer method and the electro-vibratory viscometer method. Determining the electrical conductivity of a metallurgical melt is also of particular importance for ensuring the quality of the elaboration and refining of the melt. The article also presents the possibilities of using and implementing computer systems that ensure the accuracy of determining the physical-chemical properties of metallurgical melts, as well as examples of the use of physical-chemical sensors and specific computer systems for determining the analyzed parameters. The specific electrical conductivity measurements of oxide melts are performed by direct methods (by contact), with Ohm’s law as a starting point. Thus, the article presents the voltmeter-ammeter method and the point method (or the zero method). The novelty of this article is the description and the use of specific methods and sensors for certain determinations (viscosity and electrical conductivity) for metallurgical melts. The motivation here is the presentation of the authors’ research in the addressed field. The article presents an original contribution of the adaptation and use of some methods for determining some physico-chemical parameters, including specific sensors, in the field of the elaboration of metal alloys, with the aim of optimizing their quality.

## 1. Introduction

The accuracy of determining the physico-chemical properties of the various metallic materials developed is of particular importance for their quality. Thus, according to [1], “Currently, coatings of materials with high quality and properties of hardness and resistance to wear, as is the case of martensitic steel, have been widely accepted both in the original manufacturing process of components subject to intense demands in relation to these properties, as well as in the process of sudden recovery or scheduled maintenance of these damaged components. 

For example, in the case of hydraulic turbines, cavitation erosion is a constant and harmful phenomenon, being responsible for large losses and damages in the electricity sector. However, the replacement of an already installed hydraulic unit would be unfeasible from a technical and economic point of view, with the costs of repairs being significant, and the biggest consequence is the stopping of the hydraulic unit for several days, to recover the eroded surfaces by cavitation [2].

According to [3], “The majority of catastrophic wheelset failures are caused by surface opening fatigue cracks in either the wheel tread or wheel inner. Since failures in railway wheelsets can cause disasters, regular inspections to check for defects in wheels and axles are mandatory. Currently, ultrasonic testing, acoustic emissions, and the eddy current testing method are regularly used to check railway wheelsets in service. Yet, in many cases, despite surface and subsurface defects of the railroad wheels developing, the defects are not clearly detected by the conventional non-destructive inspection system”.

As stated in [4], “One important characteristic of metallic surfaces is their roughness. In many application fields, surface roughness measurements can be a critical issue both for the maintenance of the performance of devices and during the design of particularly sensitive systems. To mention a few examples, the performance of electrochemical electrodes, which is in fact directly dependent on their roughness coefficient”.

According to [5], “Steel is one of the most basic ingredients, which plays an important role in the machinery industry. However, the steel surface defects heavily affect its quality. The demand for surface defect detectors draws much attention from researchers all over the world. However, there are still some drawbacks, e.g., the dataset is limited accessible or small-scale public, and related works focus on developing models but do not deeply take into account real-time applications. In this paper, we investigate the feasibility of applying stage-of-the-art deep learning methods based on YOLO models as real-time steel surface defect detectors”.

An important property for quantifying the quality of metallurgical melts is viscosity, which is defined as the property of a fluid to oppose the relative movement of its constituent particles. Another definition of viscosity is the property of fluids to oppose deformations that do not constitute reductions in their volume, through the development of unitary efforts. The most specific are the tangential efforts, and they develop between fluid layers in relative motion according to Newton’s hypothesis.

The viscosity of a metallurgical melt can be calculated according to the following relations [6,7,8,9]:(1)$$F=A*\;\mu \frac{v}{n}$$
or
(2)$$\tau =\mu *\;\frac{dv}{dn}$$
where:

*F*—the force that moves the top plate,

*A*—the surface of the plate,

τ—tangential unit effort,

μ—dynamic viscosity,

$\frac{dv}{dn}$—the variation of the speed perpendicular to the flow direction.

Figure 1 shows Newton’s experiment for viscosity.

To determine the viscosity of metallurgical oxide melts, two types of viscometers are mainly used: rotary and electro-vibratory.

The rotary viscometer method is based on the friction that occurs between an immersed cylinder and the respective melt, in two versions: In one, the viscosity is determined based on the calculation of the logarithmic decrement of the damping of the rotating oscillations of the cylinder immersed in the slag. In the other version, the determination is made with a viscometer where the cylinder immersion is rotated with the help of a synchronous motor.

The shortcoming of these viscometers is that they are not sensitive enough and do not yield satisfactory results for slags from loosening or refining alloys (with low viscosity). Thus, for values lower than 0.4 Ns/m^2^, the indications of the device are not constant and therefore the accuracy is lower. For this reason, in terms of the viscosity measurement for any melt, whether metallic or oxidic, the electro-vibratory viscometer was used.

The electro-vibratory viscometer method has been established in the determination of viscosity in a wide range of values, and especially for measuring in the range of 0.002 to 2 Ns/m^2^, specific to both oxide and metal melts. The vibrating viscometer method is based on the determination of the variation of the parameters of the forced oscillations of a flat plate when it is introduced into a viscous medium.

The amplitude depends on both the dimensionless viscosity and the dimensionless frequency, so there are different ways of determining the viscosity according to the measured size of the amplitude of the oscillation of the suspended system.

The functional characteristics of the electrical installations that use aggregates for the production and refining of steel, as well as the consumption of electrical energy, depend to a large extent on the electrical conductivity, and respectively the resistivity, of the materials that are found at a given moment in the furnace tank [10,11,12].

In the case of the production of ferrous alloys, the electrical conductivity decisively influences the way electricity is transformed into heat, as well as the physico-chemical processes that take place in the reaction zone.

Electrical conductivity is one of the determining physico-chemical properties in the development of the typical processes of steel production in electric arc furnaces. Additionally, the electrical conductivity of the slag for the electrical remelting under the steel slag (R.E.Z.) is very important because the electrical efficiency of the remelting installation depends on it. The slag here is a resistive medium that produces steel melting through the Joule–Lenz effect.

The electrical conductivity of a material is defined as the ratio between the density of the electric current, J, produced by placing the material in the electric field, E:J = σ E (3)

There are materials where the electrical conductivity is anisotropic, and the size and orientation of the vector J depends on the size and orientation of the vector E, in which case the electrical conductivity must be expressed by a tensor of rank 2 (a 3 × 3 matrix). Materials with a layered structure, such as some sedimentary rocks, have such a property; in their case, the conductivity in the plane of the layers can be different from the conductivity in the perpendicular direction.

In alternating electric fields, the electrical conductivity is expressed by a complex number (or a tensor of complex numbers if the material is anisotropic), called electrical admittance. In this case, the real part of the admissivity is called conductivity and the imaginary part is called susceptibility. Similarly, the quantity called admittance corresponds to conductance in an alternating field, which is the inverse of electrical impedance.

To perform electrical conductivity determinations, the conductivity cell is placed in a Wheatstone bridge assembly. To eliminate the effects of the processes that may occur at the electrodes, the measurements are performed using an alternating current.

The principal diagram of the method of determining the electrical conductivity is presented in Figure 2.

The sensors (sensitive elements—ES) used to measure the main variables related to metallurgical melts can be from the following categories:
For temperature measurement:(a)Thermocouples:(1)Radiation and convection,(2)Contact,(3)Pyrometers.(b)Thermal resistances:(1)Conductors,(2)Semiconductors (Th).(c)Pyrometers:(1)Total radiation,(2)Color (accuracy).For pressure measurement:(a)Elastic elements (manometers):(1)With a curved tube (Bourdon),(2)With a membrane,(3)With a capsule,(4)With bellows.(b)Sensitive elements with a “U” tube (U pressure gauge with liquid) (simplicity, precision):(1)With H_2_O (mm CA),(2)With Hg (1 mmHg = 1 torr; 1 physical atm = 760 torr).For measuring the flow of energetic fluids (by the constant pressure drop method and the variable pressure drop method):(a)Diaphragm discs (Kent diaphragm),(b)Nozzles,(c)Venturi tubes,(d)Rotameters.

The motivation of this article is the presentation of the authors’ research in the addressed field. The authors present their original contributions of the adaptation and use of some methods for determining some physico-chemical parameters, including specific sensors, in the field of the elaboration of metal alloys, with the aim of optimizing their quality.

The structure of the article is as follows, in sections: (1) Introduction, (2) Materials and Methods, (2.1.) Viscosity Determination (Methods for Viscosity Determination), (2.2.) Electrical Conductivity Determination, (3) Results, and (4) Conclusions. 

## 2. Materials and Methods

### 2.1. Viscosity Determination

The precise determination of the viscosity leads to the creation of optimal dispersion media in the case of using foaming slag melting processes in arc furnaces. Keeping the viscosity at low levels is recommended in the case of slags used in the continuous casting of steel, paying special attention to the lubrication of the walls of the crystallizer.

Additionally, the precise determination of the viscosity is necessary in the case of slags used in the electric remelting process under the slag (R.E.Z.).

#### Methods for Viscosity Determination

For the determination of low viscosities, from hundredths to a few poises, i.e., the range in which the viscosity of slags is found, the most suitable option is the one in which the oscillation frequency is chosen so that the amplitude of the oscillation is the maximum. The measured parameter, according to which the viscosity is determined, will in this case be the amplitude of the system oscillation.

For small viscosity values, with this method, the influence of the walls of the crucible in which the slag is located, whose viscosity is determined, can be neglected. After its calculation, even for distances of the plate from the wall of 0.2 mm, the accuracy of the determination is not influenced. This offers the possibility of building viscometers that work with small volumes of liquids to be analyzed, which is a special advantage.

The peculiarity of the electro-vibratory viscometer consists in the fact that the sensitivity is higher the lower the measured viscosity. That is why it is the most suitable device for measuring the viscosity in the previously mentioned range (0.01–2 Ns/m^2^) of slags (oxidic melts) having, under furnace conditions, a viscosity of 0.2–8 poise (0.02–0.8 Ns/m^2^).

For the study of the viscosity of oxide melts, the two methods complement each other: the first (rotary) can be used to measure high viscosities (over 1 Ns/m^2^), and the second (vibration) to measure low viscosities.

Since any study refers to conditions as close as possible to those existing in industrial metallurgical aggregates, it is sufficient to use only the viscometer with vibrations, because its precision in this field is greater.

Figure 3 shows the diagram of the electro-vibratory viscometer.

### 2.2. Electrical Conductivity Determination

#### 2.2.1. Methods for Specific Electrical Conductivity Determination

Specific electrical conductivity measurements on oxide melts are performed by direct methods (by contact), based on Ohm’s law, according to which the resistivity is:ρ = R × S/l* [ϖm](4)
where:

R is the electrical resistance,

S is the resistor section,

l is the length of the resistor.

Specific electrical conductivity is the inverse of resistivity:λ = 1/ρ [m^−1^] (5)

Experimentally, the resistance, R, of the oxide melt contained between the electrodes is determined. Mounts based on the voltmeter-ammeter method or mounts with Wheatstone or Kohlrausch points are used for this purpose.

(a)The voltmeter-ammeter method is based on Ohm’s law:
U = I × R [V](6)
and consists in measuring the current, I_x_, and the voltage drop, U_x_, in the resistance to be measured, R_x_ (melt-slag), and then calculating the resistance of the slag:R_x_ = U_x_/I_x_ [ϖ](7)

Measurement error with this method is systematic and determined by the ratio between the resistance to be measured and the internal resistance values of the devices used.

(b)The point method (or the zero method) has high sensitivity, not being influenced by the zero device or the power source.

Wheatstone, Kohlrausch, or other bridge variants and a voltage divider were used for the measurements. Figure 4 shows the diagram of the installation.

The installation for determining the electrical conductivity of oxide melts includes the following main components: the melting unit, the measuring cell, the regulation and measurement installation, and the interface between the computer and the process.

The melting unit used was a Tamman classic furnace. 

The measuring cell was made up of two tungsten electrodes with a diameter of 2 mm, which were inserted inside some alumina tubes, to protect them against oxidation. 

The assembly of the two electrodes was fixed in a device for their vertical movement and precise measurement of the immersion depth.

#### 2.2.2. Regulation and Measurement Installation 

The installation included: a frequency generator, with frequency limits between 1 Hz and 1 MHz, a power supply, with a direct current of 0–24 V/0.8 A, an alternating current amplifier, 2 alternating current millivoltmeters, E 402, and an oscilloscope, r 104.

Instead of a milliammeter, a millivoltmeter was used, which measures the voltage drop on a standard resistance, Ret = 3.5 Ω.

This immediately resulted in the intensity of the current in the circuit: I_c_ = U_mV_/R_et_ [A] (8)
where:

Ic—current intensity in the measuring circuit,

UmV (URet)—the voltage measured with the millivoltmeter,

Ret—standard resistance.

## 3. Results

### 3.1. Hardware and Software Interface

The main components of the interface are the same as in the case of temperature or surface tension determination.

The use of this interface is possible with the help of the millivoltmeter, which converts the alternating input signal into a continuous output signal. 

The level of this continuous signal was <1.5 V, within the allowed input limits in the acquisition board.

The software interface [13,14], as well as how to use it, will be presented later.

Figure 5 shows the calibration curve, η = f (U). The calibration curve has the role of ensuring the accuracy and correctness of the determinations.

### 3.2. The Stages of the Process of Determining the Viscosity of Metallurgical Melts

Before starting the determination, the viscometer was calibrated by using liquids with known viscosities, according to Table 1.

The viscosity of the slag was determined as follows: The 100 g pellet from the studied slag was inserted into the graphite crucible, and a neutral atmosphere was created at the bottom by introducing very high purity Ar, at a rate of 1.1 L/min. The probe must have a strictly vertical position in the center of the crucible.

After fixing the probe at a distance of 5–10 mm above the slag level, the electric circuit was connected to the mains and the free oscillation of the probe. With the help of fine-tuning potentiometers, the resonance amplitude of the oscillation was established, which can be appreciated by the maximum indication of the indicator of the microparameter, for a given voltage.

This indication of the measuring device (the “zero” point—allowed for calibration) remained constant during the entire experiment, with its correction being performed by varying the resistance in the coil circuit.

While it vibrates, the probe approaches the surface of the slag; at the moment of touching it, the micrometer indication decreases, which makes it possible to know the level of the slag surface according to the dimensions of the millimeter scale fixed on the stand, after which the probe is immersed in the slag at a depth of 20 mm.

-If you work manually, read the indication of the millivoltmeter, and with the help of the calibration curve, determine the viscosity value.-If working automatically, the computer instantly provides the viscosity value.

The determination of slag viscosity starts at a temperature of 1600–1620 °C and continues by lowering the temperature at a rate of 5–6 degrees/min, every 20–30 °C. After the experiments with the respective slag, a check of the stability of the calibration curve of the viscometer needs to be performed.

The accuracy of the measurements was ±5% at viscosities higher than 0.10 Ns/m^2^, with the sensitivity being higher the lower the viscosity.

Table 2 shows the characteristics of the temperature sensors used.

Figure 6a–e show images of the temperature sensors used in the conducted research.

### 3.3. Software Interface Dedicated to Determining the Viscosity of Metallurgical Melts

To launch the subprogram dedicated to the determination of the viscosity of metallurgical melts, the following steps are necessary: running the manager program, selecting the option “MEASUREMENT AREAS” from the main menu, and the selecting the “VISCOSITY” option.

The software interface includes two important areas: the control panel and the calibration curve.

The control panel in turn includes four areas: the calibration tool, the resonance level, setting the acquisition board, and experimental results, from start to exit.

The calibration type can be automatic or manual.

If the automatic type is chosen, the program has its own calibration curve, which is obtained by processing a set of values (Table 3) with statistical programs, obtained with the help of standard liquids.

After processing with the least squares method, a calibration curve of the type was obtained:η = e^(a+b*U)^ sau ln(η) = a + b*U(9)
where:

η—melt viscosity,

U—the voltage measured by the millivoltmeter.

The calculated values of the coefficients were: a = 8.76055226 and b = −0.0061427782.

If the variation of the viscometer was artificially blocked (by hand), the indication of the millivoltmeter was equal to 260, which according to the calibration equation corresponds to a viscosity of 1291 cP.

This value is considered the upper limit of the viscosity that can be measured with the presented installation. 

Table 3 shows the values of the determined viscosity and those of the standard.

Figure 7 shows the calibration values for viscosity.

If you do not want to use the automatic calibration, and implicitly the curve provided by the computer, you can select the manual calibration mode, which involves entering the numbers of your own determinations, and then their values.

Using the new pairs of viscosity values entered as an indication, the calculator determines, using the least squares method, the new calibration curve. 

This curve is only valid for the current work session, and restarting the program leads to the loss of new calibrations.

In the case of both the automatic method and the manual method, the calibration curve is drawn in the window intended for the diagram, on the left of the screen.

If between determinations, the electro-vibratory viscometer is left to freely oscillate in the air, it does not return to the initial resonant frequency, and the subsequent results will be altered by systematic measurement errors. 

This can be avoided using the resonance level option in the control panel. Any change made to the resonance level leads to the automatic recalculation by the program of the coefficients of the calibration equation, and those involved in any other calibration curve.

The settings of the acquisition board assume the establishment of the acquisition channels intended for the determination of the viscosity, and simultaneously the temperature. Program activation is performed with the START button.

### 3.4. The Hardware and Software Interface between the Computer and the Process

The main components of the hardware interface are the same as in the case of temperature, surface tension, or viscosity determination. 

The use of this interface is possible with the help of millivoltmeters, which convert the alternating input signal into a continuous output signal.

The values obtained in this way represent the Ucell and Uret, which allow the determination of the resistance of the cell, and implicitly, of the conductivity of the immersion melt:λ = K/R_CEL_ = K*I_CEL_ = K*U_RET_/R_ET_*U_CEL_
(10)

The software interface, as well as how to use it, will be presented later.

### 3.5. Calibration of the Installation

Before starting the determinations on melts, the constant of the cell was determined, and its calibration was performed.

The cell constant was determined according to the constructive parameters of the cell and experimentally with the help of a standard solution and a melt.

Depending on the constructive parameters: for the electrode–electrode-type cell, K was calculated with the relation:K = (ln2D/d)π · 1(11)

For values D = 10 mm, D = 2 mm, 1_1_ = 10 mm, and 1_2_ = 15 mm, the following constant values were obtained:K_1_ = 0.73293561138196 cm^−1^ (1_1_ = 1 cm)(12)
K_2_ = 0.48862374092130 cm^−1^ (1_2_ = 1.5 cm)(13)

The standard solution chosen was a saturated NaCl solution at a temperature of 24 °C. Its conductivity was 0.2462 Ω^−1^m^−1^. 

The results of the measurements for the saturated NaCl solution are presented in Table 4.

The difference between the values obtained for the electrical conductivity with the help of the geometric constant and the theoretical conductivity of NaCl (0.2462 Ω^−1^cm^−1^) is explained by the fact that the geometric constant does not take into account the resistance of the connecting conductors and the electrodes.

Experimental: To eliminate this error, the constant, K, was experimentally determined, based on the electrical conductivity of the standard solution:K = (λ*U)/I(14)

The frequency at which the polarization resistance is minimal was considered 20 KHz.

In this case, a constant of the cell resulted as K = 00.67295 cm^−1^, and this constant can be used for a set of experiments.

In order to verify the calibration of the measuring cell, determinations were performed on melts of CaCl_2_ at 1000 °C.

The results are presented in Table 5.

Comparing the average value of the obtained experimental results, λ_med_ = 2.71096 Ω^−1^ cm^−1^, with the obtained value from the specialized literature, λ = 2.69 Ω^−1^ cm^−1^, it can be said that the calibration of the installation was successfully carried out.

The difference between the dimensional constant, K_d_ = 0.73293561138196 cm^−1^, and the experimental constant, K_e_ = 0.67295 cm^−1^, was ΔK 0.061 cm^−1^, and their ratio was:
K_d_/K_e_ = 1.09(15)

## 4. Conclusions

The study and determination of the physico-chemical properties related to metallurgical melts are particularly important for quality assurance in their development. In this context, the viscosity of metallurgical melts is of great importance in the elaboration. This importance consists in ensuring the optimal mass and thermal transfer conditions between the atmosphere of the processing aggregate, the additive materials, and the processed melt.

Among the viscosity determination methods, the relevant ones are the rotary viscometer method and the electro-vibratory viscometer method.

Additionally, electrical conductivity is one of the determining physico-chemical properties in the development of the typical processes of steel production in electric arc furnaces.

The software interfaces specific to the determination of the specific physico-chemical properties of metallurgical melts ensure the necessary accuracy of these determinations.

The correct use of sensors, both for temperature and for viscosity, is particularly important for the accuracy of the determinations.

The originality of this article consists in the conception and adaptation of some methods for determining some physical-chemical parameters (viscosity and electrical conductivity) for metallurgical melts, to increase the quality of their production.

For future research directions, the authors have in mind the conception, realization, and implementation of some software products which, based on the determination of physical-chemical parameters, lead to a higher level of optimization of the quality of metallurgical melts during elaboration.

## Figures and Tables

**Figure 1 sensors-23-03965-f001:**
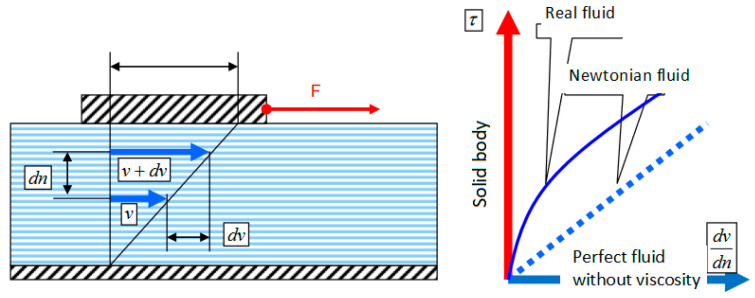
Newton’s experiment for viscosity.

**Figure 2 sensors-23-03965-f002:**
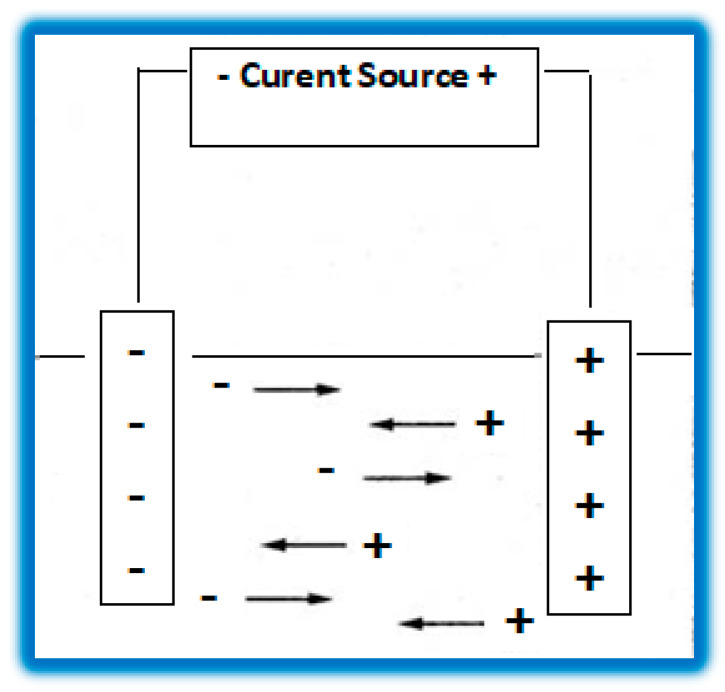
The principal diagram of the method of determining the electrical conductivity.

**Figure 3 sensors-23-03965-f003:**
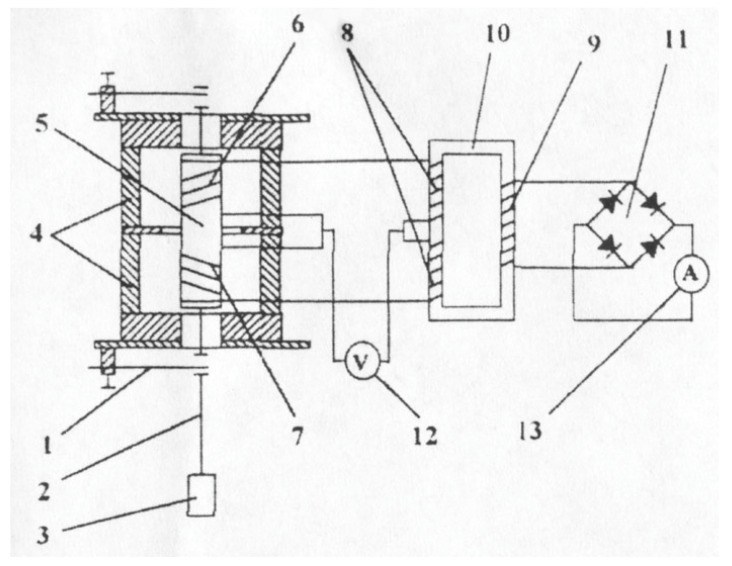
Diagram of the electro-vibratory viscometer. 1—flat springs; 2—molybdenum rod; 3—molybdenum probe; 4—permanent magnets; mild steel core; 6, 7—coils; 8—primary windings, 9—secondary windings; 10—differential transformer; 11—recovery bridge; 12—millivoltmeter; 13—ampermeter.

**Figure 4 sensors-23-03965-f004:**
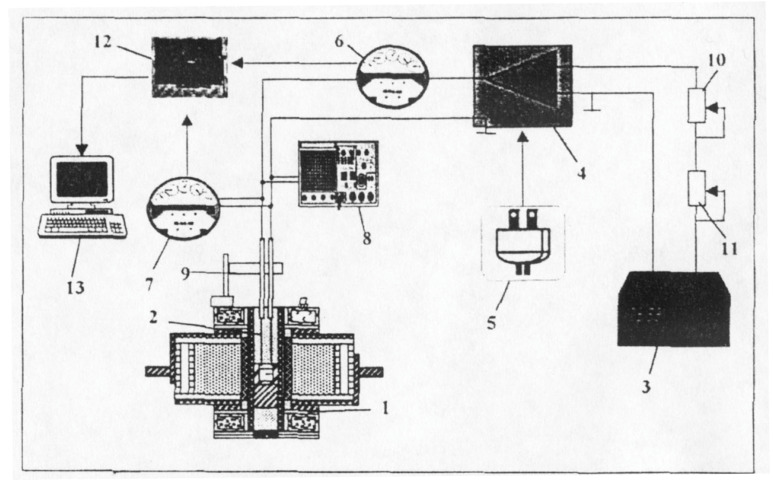
Scheme of the installation for determining electrical conductivity. 1—Tamman oven; 2—Tungsten electrodes; 3—de-frequency generator; 4—alternating current amplifier; 5—DC power supply; 6—alternating current milliammeter; 7—alternating current millivoltmeter; 8—oscilloscope; 9—device for measuring the immersion depth of the electrodes; 10—fine adjustment potentiometer; 11—gross adjustment potentiometer; 12—signal acquisition board; 13—computer.

**Figure 5 sensors-23-03965-f005:**
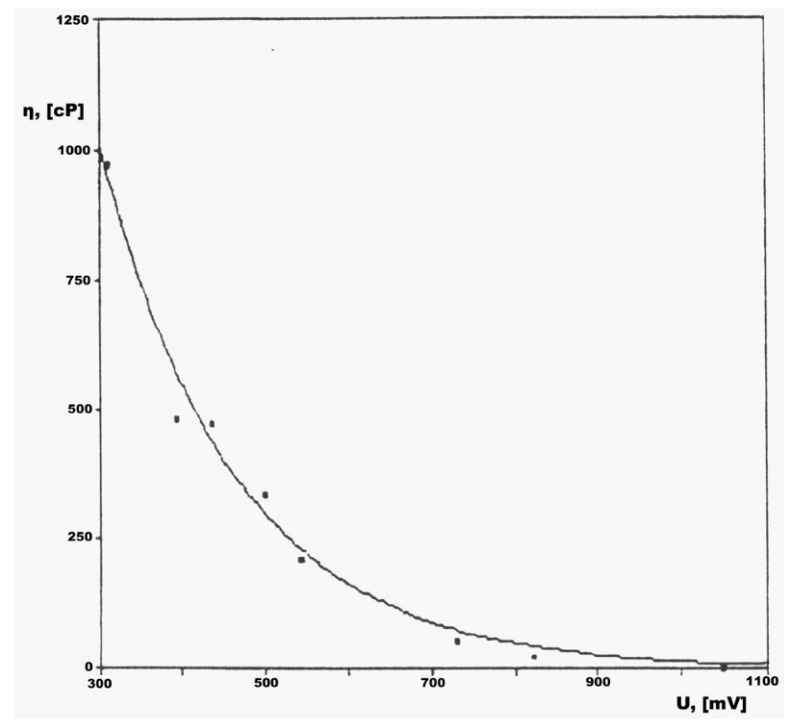
Calibration curve, η = f (U).

**Figure 6 sensors-23-03965-f006:**
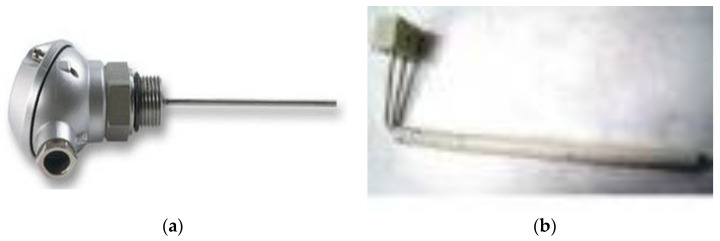
Temperature sensors used. (**a**) J-type sensor (Fe–Ct). (**b**) S-type sensor (Pt-Rh). (**c**) Sensor type PT100 sheath 150 mm-50/+400C 0628-0033-0.5. (**d**) Sensor type PT100B-6X150-A304-0000-3.0TTS. (**e**) Sensor type PT100B-6X100-MGO-0000-DIN/B.

**Figure 7 sensors-23-03965-f007:**
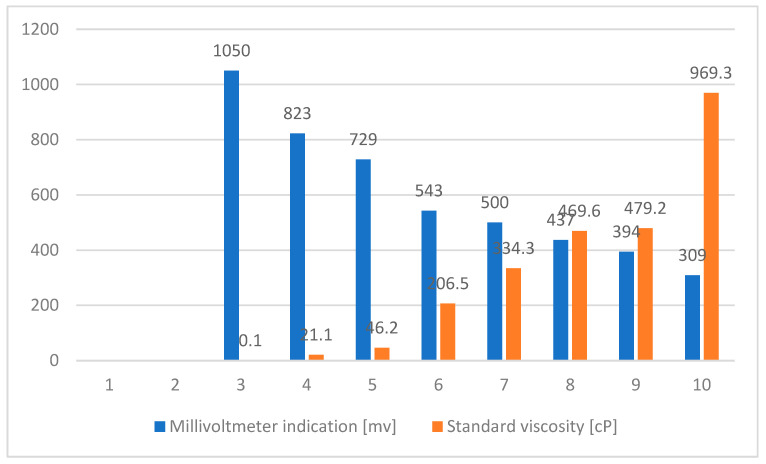
Calibration values for viscosity.

**Table 1 sensors-23-03965-t001:** Viscosity of some standard liquids.

Current Number	The Standard Liquid	Viscosity		Temperature, °C
		Poise	Ns/m	
1	Distilled water	0.01	0.001	25
2	Butyl phthalate *	0.16288	0.016288	25
3	Water—glycerin	0.45724	0.045724	25
4	Silicone 200 *	2.0065	0.2065	25
5	Standard liquid *	3.3438	0.334338	25
6	Standard liquid *	4.7925	0.47925	25

* National Institute of Metrology.

**Table 2 sensors-23-03965-t002:** The characteristics of the temperature sensors used.

Characteristic		Sensor Type			
		Iron-Constantan	Cromel-Alumel	Pt-PtRh 10%	Pt-PtRh 13%
Maximum temperature of use, °C	Continuously	500	$650\ldots 1000^{\;}$	$1100\ldots 1300{\;}^{}$	$1100\ldots 1400^{}$
	Flashing	500	$850\ldots 1200{\;}^{}$	$1200\ldots 1600{\;}^{}$	$1200\ldots 1600^{}$
Protective sheath		Carbon steel Stainless steel	Carbon steel Stainless steel Refractory steel	Ceramic Refractory steel	Ceramic Refractory steel
Maximum nominal length(mm)		2000	2000	500; 1000; 2000	500; 1000; 2000
Minimum nominal length (mm)		250; 500	250; 500	500; 1000	500; 1000
Minimum immersion length (mm)		150; 220	150; 220; 250; 300	150; 250; 300	150; 250; 300

**Table 3 sensors-23-03965-t003:** Viscosity values measured and given by the standard.

Millivoltmeter Indication (mv)	Standard Viscosity (cP)
1050	0.1 *
823	21.141
729	49.203
543	206.5
500	334.38
437	469.61
394	479.25
309	969.37

* The theoretical value for free oscillation in air.

**Table 4 sensors-23-03965-t004:** Experimental parameters.

V (KHz)	h (mm)	I (A)	U (V)	K (cm^−1^)	λ (Ω^−1^m^−1^)
20	10	0.3	0.82	0.7329	0.26813
20	15	0.3	0.56	0.4886	0.26175

**Table 5 sensors-23-03965-t005:** Electrical conductivity of CaCl_2_ at 1000 °C.

V (KHz)	I (A)	U (V)	λ (Ω^−1^m^−1^)
20	0.3	0.076	2.65638
20	0.3	0.073	2.76554

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
