# Peer review of "Possibilities of Physical-Chemical Sensors’ Use for Optimizing the Processing of Metallurgical Melting Based on Computer Systems"

_sensors, 2023, doi:10.3390/s23083965_

Round 1
Reviewer 1 Report (Previous Reviewer 2)
After the re-submission, I find the paper has been improved. However, I do not agree with the quality of Figures 2, 3, 4, 5 and 6. They are quite old copy-pasted images, and/or only pictures. This must be changed to be more rigorous and scientific. This kind of issues are not in line with the prestige of the Journal. Also, the results as presented lack scientific soundness. Authors should provide a serious major revision to the manuscript for its consideration in Sensors.
Author Response
We restored the quality of the reported figures.
Reviewer 2 Report (Previous Reviewer 1)
The authors have improved the quality of this article. From my aspect, it might be good enough to be published.
Author Response
Thank you for your appreciations! I'm honored!
Prof. Adrian IOANA
This manuscript is a resubmission of an earlier submission. The following is a list of the peer review reports and author responses from that submission.
Round 1
Reviewer 1 Report
The authors discussed the possibilities of optimizing the quality of the elaboration of metallurgical melts by determining their physical-chemical properties. This article seems to be more like a technical report instead of a research article. I suggest that it needs to be a throughout polish and rearrangement.
1. Line 22, “according to [1], "Currently, coatings…”, It is the first time for me to read the quotation using this way. I am not very sure whether it is suitable or not. The same situations happen in the line 34, line 41 and line 46.
2. At the end of Part 1, please add the motivation of this manuscript as well as summarize the shortcomings of previous studies on this topic.
3. At the end of Part 1, please provide a brief introduction to the paper’s arrangement for each part.
4. The novelty of the article needs to proposed and highlighted. However, I am afraid that I can not find it in the current form of this article.
5. Abstract, please add what you find and the significance of this research.
6. Part 4, please give more quantitative conclusions if possible.
Thus, I suppose this article needs to be rewritten and sharpen from the beginning to the end extensively. I encourage the authors to rewrite it and submit it again.
Reviewer 2 Report
General comment:
This manuscript presents development aspects to optimize the quality of metallurgical melts by measuring physical-chemical properties. The paper analyzes two specific properties, viscosity, and electrical conductivity. The work is relevant in the field of sensors and measurements for optimizing fabrication processes. Though the proposal is well-motivated in its area of study, the manuscript lacks scientific soundness. The methodological framework is not clear, and the results and conclusions are not well supported. The manuscript is weak and should be highly enhanced before it can be considered for the journal. I have some points that should be addressed.
Comment 1:
In the Abstract, the authors explain briefly two methods for measuring the viscosity; however, there is no mention of the method(s) for measuring the electrical conductivity.
Comment 2:
In the Introduction, there is not a diagram or schematic which illustrates the measurement of electrical conductivity.
Comment 3:
In the Introduction (pages 3 and 4). Why it is important to give a list of sensors which are not within the scope of the paper?, in terms of viscosity and electrical conductivity.
Comment 4:
Diagram in Fig. 2 is awful. This kind of issues make the paper to lose seriousness.
Comment 5:
Same comment as in the previous, for Fig. 2
Comment 6:
The authors should limit the background to the specific method used in the work. In the case of electrical conductivity, they refer to bridge circuits, voltage dividers, etc., which is quite general. Please be specific.
Comment 7:
The calibration curve of Fig. 4 is not discussed nor explained in the text. Also, its quality is not good.
Comment 8:
Authors should enhance the quality of Fig.5. Also, a thorough discussion is needed.
Comment 9:
Why is important to measure electrical conductivity by using an alternating current excitation? There is no explanation about this issue.
Comment 10:
The conclusions of the paper are vague. They do not give a concise statement of the findings derived from the work. Moreover, they are limited to give (weak) qualitative ideas.